# The Effect of Non-Uniform Magnetic Field on the Efficiency of Mixing in Droplet-Based Microfluidics: A Numerical Investigation

**DOI:** 10.3390/mi13101661

**Published:** 2022-10-02

**Authors:** Masoud Rezaeian, Moein Nouri, Mojtaba Hassani-Gangaraj, Amir Shamloo, Rohollah Nasiri

**Affiliations:** 1Department of Mechanical Engineering, Sharif University of Technology, Tehran 11365-8639, Iran; 2Department of Protein Science, Division of Nanobiotechnology, KTH Royal Institute of Technology, 171 65 Solna, Sweden

**Keywords:** microfluidics, droplet-based mixing, ferrofluids, non-uniform magnetic field, droplet

## Abstract

Achieving high efficiency and throughput in droplet-based mixing over a small characteristic length, such as microfluidic channels, is one of the crucial parameters in Lab-on-a-Chip (LOC) applications. One solution to achieve efficient mixing is to use active mixers in which an external power source is utilized to mix two fluids. One of these active methods is magnetic micromixers using ferrofluid. In this technique, magnetic nanoparticles are used to make one phase responsive to magnetic force, and then by applying a magnetic field, two fluid phases, one of which is magneto-responsive, will sufficiently mix. In this study, we investigated the effect of the magnetic field’s characteristics on the efficiency of the mixing process inside droplets. When different concentrations of ferrofluids are affected by a constant magnetic field, there is no significant change in mixing efficiency. As the magnetic field intensifies, the magnetic force makes the circulation flow inside the droplet asymmetric, leading to chaotic advection, which creates a flow that increases the mixing efficiency. The results show that the use of magnetic fields is an effective method to enhance the mixing efficiency within droplets, and the efficiency of mixing increases from 65.4 to 86.1% by increasing the magnetic field intensity from 0 to 90 mT. Besides that, the effect of ferrofluid’s concentration on the mixing efficiency is studied. It is shown that when the concentration of the ferrofluid changes from 0 to 0.6 mol/m^3^, the mixing efficiency increases considerably. It is also shown that by changing the intensity of the magnetic field, the mixing efficiency increases by about 11%.

## 1. Introduction

Microfluidic systems have a high potential to be used in biotechnological and pharmaceutical applications due to small sample consumption and rapid response [1,2,3]. The mixing process plays a crucial role in different physical processes in microchips such as heat transfer, mass transfer, and chemical reactions [4,5,6]. Due to the low Reynolds number and laminar flow regime in the microchannels, achieving uniform mixing in a short time interval has always been a challenging issue in microfluidic technology [7,8]. Inefficient mixing is not appropriate for fast analysis operations such as Polymerase Chain Reaction (PCR) and cell lysis [9,10]. It also creates a heterogeneous mixture that reduces detection accuracy and production quality [11,12,13]. For these reasons, rapid mixing and achieving high mixing efficiency are essential in lab-on-a-chip platforms [14].

Based on the fluid state within the micromixers, the mixing process can be classified into single-phase mixing and droplet-based mixing [15]. In single-phase mixing, the mixing efficiency is limited by the diffusion flux, and the dispersion of the solute elements along the channel is high, so the efficiency is low [16,17]. Droplet-based mixing is another approach in which droplets of a dispersed phase that usually contains target samples form in a continuous phase [18]. This technique partially overcomes the flaws mentioned for single-phase mixing. In this method, the two immiscible phases include the continuous phase, the medium in which the droplets form, and the dispersed phase, the droplet phase, and internal circulation and isolated medium increase the mixing efficiency within droplets [2,14]. Moreover, samples in droplets are protected from contaminants of the outer environment [19].

Rapid mixing inside the droplet occurs based on convection and diffusion [20]. However, convection in straight channels is difficult to achieve due to the symmetric internal circulation of the two currents in each half of the droplets. To resolve this issue, Yang et al. [21] integrated a T-junction with a serpentine channel where the stretching and folding of entire droplets resulted in the development of chaotic advection. The part of the droplet that had the effect of the inner arc made a small rotation, while the other side produced a large one. The asymmetric circulation flow allowed mass transfer in both axial and radial directions and ultimately increased mixing efficiency [21].

Kushid et al. used Computational Fluid Dynamics (CFD) and Particle Image Velocity (PIV) techniques and investigated the rotation of the fluid within droplets. They showed that the flow rate has little effect on the internal circulation when the droplet length is sufficiently large [22]. By adding a scalar equation to the Volume of Fluid (VOF) methodology, Chandorkar et al. studied the changes in the concentration of the microdroplets and the multiphase flow mixing process under different operating conditions [23]. To further study the mixing process in the serpentine channel, the effect of dimensional parameters on mixing was investigated. It was observed that the ratio of droplet size to channel width, the flow rate of dispersed phase to continuous phase, and the capillary number ($\mathrm{Ca}=\mu_{c}V_{c}/\sigma )$, representing viscous forces versus surface tension, were the most critical factors affecting the mixing process. In contrast, the Reynolds number ($Re=\rho V_{c}l/\mu_{c})$, which is the ratio of inertial to viscous forces, had a minor effect on the mixing index [24]. The investigation of the internal mixing behavior of the droplet flow in the serpentine channel was evaluated and it was reported that the best mixing occurs when the droplet size is comparable to the channel width. To study the chemical reactions in the droplet, Madadelahi et al. examined both the mixing and the chemical reactions within the dispersed phase and the effect of the secondary flow. They observed that the chemical reaction inside a droplet started from its front regions [15]. Acting to change the velocity field inside the droplets given Dean vortices, channels with complex geometries have a competitive edge over simple straight channels to make the mixing process occur promptly [25]. Yu et al., using an objective generic algorithm, designed an optimal sinusoidal micromixer, considering two decisive parameters: the average concentration difference within the droplet and pressure drop along the channel. With the Dean vortices and contraction–expansion cross-section intensifying velocity gradients, setting the stage for asymmetric circulating flow within the droplets, this optimized channel would enhance the mixing efficiency considerably [26].

In the active mixers, the development of chaotic flow is complicated. Due to the advantages of magnetic force such as remote control, easy integration, and comfortable fabrication, its use is an excellent way to increase mixing. The combination of the magnetic field with the magnetic fluid flow, ferrofluid, resulted in a micromagnetofluidic field [27].

Several studies have been performed to investigate the droplet formation and the mixing within droplets using magnetic particles [28]. Some of these works evaluated the mixing efficiency and merging of microparticles into a magnetic platform [29]. However, because these platforms provide manipulation conditions for a single droplet, they are not suitable for processes such as drug screening, which require high-throughput droplet formation [30]. Magnetic stirring inside the droplet is an alternative way to enhance the mixing. Magnetic polymer beads [31], polymer structures containing magnetic nanoparticles, and magnetic barriers were used to stir within the droplet. In this technique, the dipole–dipole interaction between the magnetic particles in the presence of the magnetic field caused a chain of particles to create the stirrer.

Maleki et al. examined the effect of a uniform magnetic field on the mixing efficiency using a heterogeneous magnetic fluid, ferrofluid. The gradient of magnetic particle concentration in the droplets led to the momentum gradient and the making of magnetic bulk forces. The induced magnetic force created chaotic advection by asymmetrical internal recirculation and increased the mixing dramatically [32].

In previous works, there is a lack of study about the effect of concentration changes of ferrofluids and intensity of non-uniform magnetic field on the mixing performance. In this study, the droplet formation and mixing efficiency within droplets in different concentrations and non-uniform magnetic field intensities were investigated in a microfluidic device with a flow-focusing unit droplet generation.

## 2. Materials and Methods

### 2.1. General Overview of the Model

The main goal of this study is to study the effect of a non-uniform magnetic field on the efficiency of mixing inside droplets. The strategy adopted in this study is as follows:

In the first step, the process of droplet formation is simulated by solving the phase-field equations. The main purpose of this step is to study the velocity field generated inside droplets after formation.

The second step is to determine the effect of the magnetic field on the fluid flows inside droplets. For this purpose, the magnetic field is calculated, and its effect on the velocity field is investigated. This way, the velocity field, which is affected by both droplet generation and the magnetic field, is calculated.

The third step is to determine how the fluid flows inside the droplet affect the mixing efficiency. To investigate this effect, a separate model of the droplet is generated. The calculated flows affected by the droplet generation process and the magnetic field are imported into this model, and then the flow inside the droplet and the process of mixing are investigated.

### 2.2. Material Parameters and Geometry

In this study, a CFD model has been used to simulate droplet formation and mixing inside droplets in the presence of a non-uniform magnetic field. The material properties were extracted from a similar experimental study to simulate these conditions. The density and viscosity of the ferrofluid are 1100 kg/m^3^ and 5 mPa·s at 25 °C. The density of the mineral oil at 25 °C is 840 kg/m^3^. The permeability of the free space and the initial susceptibility of the ferrofluid are 4$\pi \times 10^{-7}\;N/A^{2}$ and χ = 0.36, respectively [33].

The geometry of the microfluidic device consists of three distinct parts. The first part is the space between the magnet and microfluidic channels, an essential region for simulating the magnetic field. The second part is a permanent magnet, and the third is the microchannels containing mineral oil and the aquatic phase. These materials are used as two immiscible phases for droplet formation. A flow-focusing technique and a permanent magnet are considered as a base platform for droplet formation and a source for the magnetic field, respectively. Figure 1 shows the geometry under which the droplet formation is simulated. It contains two 175 µm inlets (inlets 1 and 2) for the continuous phase, one 120 µm inlet (inlet 3) for the dispersed phase, and one outlet with 300 µm width (outlet).

The mesh independence study showed that the droplet size is independent of the mesh size with a grid density of 20 divisions per 100 μm. The mapped mesh was used inside microchannels and the free tetrahedral in other domains.

### 2.3. Governing Equations

#### 2.3.1. Droplet Formation

The governing equations for droplet generation are the conservation of mass (continuity), Equation (1), and momentum, Equation (2) [34,35].
(1)∇⋅u=0 
(2)$$\rho \frac{\partial u}{\partial t}+\rho \left(u\cdot \nabla \right)u=-\nabla p+\eta \nabla^{2}u+F_{s}+F_{m}$$

In this equation, ρ, P, and η represent the fluid density, the pressure, and the dynamic viscosity, respectively. F_s_ is the force caused by the surface tension, and $F_{m}$ is the magnetic volume force. Surface tension body force ($F_{s}$) can be obtained using phase-field equations [36]: (3)$$\frac{\partial \varphi }{\partial t}+u\cdot \nabla \varphi =\nabla \cdot \frac{\gamma \lambda }{\varepsilon^{2}{}_{\mathrm{pf}}}\nabla \psi ,$$
where φ represents the phase-field variable which is used to model the immiscible interaction fluids and govern their interface action. In this method, the phase-field variable φ determines the distinct phases and describes a smooth transition across the interface from one to zero. The conservative level set equation is defined as follows:(4)$$\psi =-\nabla \cdot \varepsilon^{2}{}_{\mathrm{pf}}\nabla \varphi +\left(\varphi^{2}-1\right)\varphi +\frac{\varepsilon^{2}{}_{\mathrm{pf}}}{\lambda }\frac{\partial f}{\partial \varphi },$$
(5)$$\lambda =\frac{3\varepsilon_{\mathrm{pf}}}{\sqrt{8}},$$
(6)$$\gamma ={\chi c}^{2}{}_{\mathrm{pf}},$$
where γ represents the reinitialization parameter and ε denotes the interface thickness where the level set function varies from 0 to 1 (core represents the reinitialization). Reinitialization is a necessary step to maintain the acknowledged distance property and control numerical stabilization. ψ is a supplemental variable to convert the fourth-order equation into two second-order equations. χ is the mobility tuning parameter. σ represents the surface tension coefficient, and ε is the interfacial thickness.

#### 2.3.2. Mixing Process

The following convective/diffusive equation expresses the non-dimensional mass transport across the fluidic domain; this equation is used to model the distribution of magnetic particles in the computational domain [37].
(7)$$\frac{\partial C}{\partial t}+u_{p}\cdot \nabla C=\nabla \cdot \left(D\nabla C\right),$$
where C denotes the dimensionless concentration, D is the inefficient diffusivity, and u_p_ is the velocity of the particles. The diffusive behavior of magnetic particles is influenced by the external magnetic field. This effect is shown by an additional velocity component given by Stokes drag law:(8)$$u_{p}=u+u_{\mathrm{Mag}}=u+\frac{F_{\mathrm{mag}}}{6{\pi \eta r}_{p}},$$
where $r_{p}$ is the magnetic particle radius and $F_{\mathrm{mag}}$ is the magnetic force on each particle given by:(9)Fmag=12Vpμ0 ⋅∇χH2,
where $V_{p}$ is the volume of any single magnetic particle, $\mu_{0}$ is the permeability of the free space, χ is the local susceptibility of the ferrofluid, and H is the magnetic field intensity. The interfacial discontinuity of concentration can be expressed as follows [37]:(10)$$c_{c}={\mathrm{kc}}_{d},$$
where k is the distribution coefficient, the indices c and d correspond to the continuous and dispersed phases, respectively. Additionally, the flux continuity condition is established at the interface.
(11)$$D_{d}\frac{\partial C_{d}}{\partial n}=D_{C}\frac{\partial C_{C}}{\partial n}$$

Due to the moving interface, it is impossible to apply a boundary condition to the interface. Kenig et al. addressed this problem in their research [37]. This method combines interfacial boundary conditions with the mass transport equation so that these conditions are satisfied only in the vicinity of the immiscible fluid interface. New source terms are added to the mass transport equation. Consequently, the following equations are reached and solved for all fluidic domains.
(12)$$\frac{\partial C_{d}}{\partial t}+u_{p}\cdot \nabla C_{d}=\nabla \cdot \left(D\nabla C_{d}\right)+\left(D_{C}\frac{\partial C_{C}}{\partial n}-D_{d}\frac{\partial C_{d}}{\partial n}\right)$$
(13)$$\frac{\partial C_{c}}{\partial t}+u_{p}\cdot \nabla C_{c}=\nabla \cdot \left(D\nabla C_{C}\right)+\alpha_{2}\left(C_{C}-\frac{C_{d}}{K}\right)$$

#### 2.3.3. Calculation of the Magnetic Force

In order to calculate the induced magnetic force on the nanoparticle, Maxwell equations should be implemented. Maxwell equations for non-conducting media are expressed as follows [38]:(14)∇×H=0,
(15)∇⋅H=0,
where H represents the magnetic field intensity and B denotes the magnetic flux density. The relation between H and B in different parts of the computational domain is defined as the following:(16)B={μ0 (H) in Ωfμ0(M+H) in ΩΩf.

In the above equation, $\mu_{0}$ is the permeability constant with the amount of 4 $\pi \times 10^{-7}$ VsAm and M represents the magnetization of the ferrofluid in the medium. $\Omega_{f}$ demonstrates the inner region of the microchannel filled with ferrofluid and ΩΩf is related to the non-magnetizable area in the outer region of the microchannel consisting of the air. Implementing the mentioned approach, we have modeled the ferrofluid behavior in the droplet domain to calculate the magnetic force presented in the momentum equation [39].

To evaluate the mixing performance, the parameter for mixing efficiency is expressed as [21]:(17)$$M_{i}=1-\frac{\int_{A}\left|C_{i}-C_{d}\right|\mathrm{dA}}{\int_{A}\left|C_{0}-C_{d}\right|\mathrm{dA}},$$
where $C_{i}$ is the profile of the concentration in the outlet and $C_{d}$ is the normalized average of all the input concentrations.

### 2.4. Numerical Simulation

A CFD model is developed to investigate the process of droplet formation and mixing inside the droplet. In the presented CFD model, all the governing equations are solved coupled with each other. In the discretization of the domain, first-order elements were used to discretize the pressure, and second-order elements were used to discretize the velocity field throughout the domain. The other parameter that is needed to be discretized is the phase-field variable. Due to its similarity to the velocity field, second-order elements were also used to discretize this variable in the domain. In addition to that, since the study is a time-dependent study, an implicit second-order Backward Differential Formula (BDF) was used to discretize variables in time.

### 2.5. Droplet Model

In this study, a finite element model of the droplet is used to investigate the mixing efficiency. In the droplet model, the interaction between the dispersed phase and the continuous flow is excluded and the focus of this model is on the flow inside droplets. Two steps are required to set up the droplet model. The first step is extracting the velocity field inside droplets after the droplet formation and the second step is importing the velocity field into the droplet model. This way, a model for further investigation of the fluid flow inside droplets is set up.

The flow inside a droplet is a closed flow, meaning that there is no inlet or outlet. Besides that, fluid inside a droplet interacts with its inner wall. The inner wall of the droplet does not impose any shear stress on the fluid; also, the fluid does not move perpendicular to the wall. The boundary conditions on the inner wall of the droplet are:(18)∇·V=0,
(19)τ=0

### 2.6. Boundary Conditions

For inlets, velocity boundary conditions are used. A pressure boundary condition is used for the outlet and a wetted wall condition is set for the walls of the microchannels. At first, the only fluid present in the channels is mineral oil, so by setting the contact angle of the wall to zero, the oil will always stay in contact with walls and this will make the walls hydrophobic so the water droplets move along the channels without any contact with the walls [40].

## 3. Results and Discussion

### 3.1. Validation

To validate our results with experimental data, the mixing efficiency and the droplet size of our simulations are compared to experimental and numerical results presented by Bai et al., shown in Table 1 [41]. The mixing efficiency of our simulation without magnetic field and L/W = 10 is 46.1% and in Bai et al.’s simulation, this parameter is about 47%. The droplet sizes in the simulation and Bai et al.’s results are 208 and 240 µm, respectively, which means a 13.3% error [41].

### 3.2. Droplet Generation

As mentioned, the dimensionless parameters governing the droplet formation are the capillary number, flow rate ratio, and viscosity ratio. In this study, at first, the microchannels are filled with the continuous phase; therefore, the walls of the microchannels become hydrophobic. Then, two immiscible phases are pumped with a specific flow rate through the channels, the dispersed phase moves forward, and the tip of the thread forms a round shape. The shear thinning process makes the water phase thinner, and a necking stage occurs in the dispersed thread. Finally, the thread is broken up by the viscous shear stress, and a droplet of the aqueous phase is formed. The fluidic part of this simulation consists of mineral oil as the continuous phase and two aqueous solutions (i.e., ferrofluids and water) as the dispersed phase, the droplet phase. Considering the width of the channel (h = 300) and the cross-section aspect ratio of 1.5, Figure 2 shows the droplet formation in different flow ratios. As can be seen in Figure 3, by increasing the ratio of the dispersed flow rate to the continuous flow rate, the droplet size increases, and the time in which a droplet forms decreases (it is defined as the frequency of droplet formation).

### 3.3. Mixing Inside Droplets

In this research, the effect of the magnetic field, as an active method, on mixing inside the droplet has been investigated.

#### 3.3.1. Mixing with Changing the Direction of the Magnet

The direction of the magnetic field was altered by changing the position of the magnets in order to compare the mixing efficiency within droplets. Figure 4 and Figure 5 show how the direction of the magnetic field affects the mixing process. When the magnetic field is aligned with the interface of two solutions in droplets, convection is not triggered inside the droplets. As a result, the mixing efficiency is low (Figure 4).

On the other hand, when the direction of the magnetic field is perpendicular to the microchannel, the convection process is triggered (Figure 5). Thus, this situation has more effect on mixing compared to the direction mentioned in Figure 4.

#### 3.3.2. Mixing Efficiency with Different Concentrations of Ferrofluids and Magnetic Field Intensities

In the next step, we investigated the effect of the concentration of ferrofluids and the magnetic field intensity on the mixing index. Figure 6 shows that when the ferrofluid concentration is lower than 0.6 mol/m^3^, changing the ferrofluid concentration significantly affects the mixing efficiency. After this specific value, any increase in the concentration does not have a noticeable effect on the mixing efficiency. Figure 6 also illustrates the effect of magnetic fields with varying intensities (from 0 to 90 mT) on the efficiency of mixing within droplets. The results show when the magnetic field increases from 10 to 90 mT, the mixing efficiency inside the droplet increases from 75 to 86.1%. Droplets exposed to a non-uniform magnetic field experience magnetic force in two ways. First, because of the distinct solubility (dispersion) of magnetic particles in two immiscible phases and concentration gradient, the magnetic force performs at their interface which causes diffusion in the interface of the two solutions. Moreover, once the miscible phases merge, a concentration gradient exists at their interface. The movement of nanoparticles in droplets gives rise to the convection which has the most impact on the internal flow patterns.

By applying the magnetic field, depending on its strength, the existing magnetic gradient at the interface of the miscible phases gives rise to two different mixing behaviors. As shown in weak magnetic fields, the dominant mode of the mixing is molecular diffusion, and the magnetic force accelerates the migration of the nanoparticles. It only improves the diffusive function of the particles, so they have little impact on mixing enhancement. As the intensity of the magnetic field increases, the magnetic force disrupts the symmetric flow within the droplet and improves the mixing efficiency by enhancing the convection.

Figure 7 shows the flows generated inside the droplet after formation when no magnetic field is applied (Figure 7a) and when the magnetic field is present (Figure 7b). As shown in Figure 7a, the flow generated inside the droplet in the absence of the magnetic field consists of two microvortices at the top and bottom of the droplet (shown with 1 and 2). There are also two relatively smaller microvortices at the upper and lower parts of the droplet’s tip (shown with 3 and 4). The other component of the flow is a uniform rightward flow in the middle part of the droplet. The vortices in the upper half of the droplet (1 and 3) are the main cause of mixing in that area and vortices in the lower half of the droplet are the main cause of mixing in the lower half of the droplet. However, mixing efficiency is high in the upper and lower parts of the droplet since there is a uniform flow in the middle part of the droplet (shown with 5). Due to this uniform flow, transport of species from the lower half to the upper half and vice versa is limited and mostly relies on diffusion.

Figure 7b shows the flow generated inside the droplet in the presence of a magnetic field. It is shown in the figure that the flow is not symmetric as in Figure 7a and it is because of the magnetic field’s effect. There are four microvortices but their position is not symmetric with respect to the droplet’s symmetry line. Two other differences should be mentioned. The first one is an additional microvortex generated in the droplet when the magnetic field is present. This microvortex is shown with 6, 7, 8, 9, 10 in Figure 7b. The rectangle in Figure 7b shows that the direction of the uniform flow in the middle part of the droplet deviates from its previous direction (shown with 5). This deviation is toward the source of the magnetic field. This deviation enhances species’ transport from the droplet’s lower half to the upper part. This way, species are transported to the upper half because of convection and diffusion. This way, the efficiency of mixing is increased when the magnetic field is present.

Blusov et al. showed that asymmetric flow inside the droplet enhances the mixing process inside the droplet [42]. For this reason, the process of mixing is simulated and the results are shown in Figure 8. This figure shows how two fluids mix as they interact. Therefore, it can be concluded that the main part of the mixing occurs at the tip of the droplet, where the microvortex generated by the magnetic field is formed.

#### 3.3.3. Active and Passive Mixers

According to the literature, integrating the serpentine channel with flow-focusing in passive mixers is an efficient way to circulate the flow within the droplet asymmetrically. To analyze the mixing in active and passive modes, we compared the mixing performance in a serpentine channel, straight ones with two different lengths, and straight channels affected by the magnetic field.

#### 3.3.4. Comparison between Passive and Active Methods

To compare the mixing efficiency in active and passive methods, we first examined each of the two modes separately. In passive methods, when the magnetic field is not applied, we first performed the mixing process in flow-focusing with different channel lengths of 6 mm and 12 mm, and the mixing efficiency was 57 and 65.4%, respectively. The results show that increasing the channel length leads to a relative enhancement in the mixing index. Then, by changing the channel geometry, we examined the mixing process within the serpentine channel and observed a significant increase in the mixing efficiency.

In the active state, the mixing process occurs in the presence of a magnetic field at the intensity of 90 mT. We investigated the effect of the change in channel length for the 6 mm and 12 mm lengths and mixing efficiency was 75.2% and 86.1%, respectively. According to these results, increasing the length of straight channels in the active mode has a more significant impact on increasing mixing efficiency than in the passive method. A comparison of all the results obtained with the passive and active mixers is shown in a histogram in Figure 9. This figure shows that in the same length of channel, the serpentine geometry has about 13% lower mixing efficiency than the platform with the 90 mT magnetic field. In addition, in terms of fabrication, a straight channel with a permanent magnet is easier to fabricate than a serpentine channel.

Figure 10 shows the mixing efficiency in different time steps for a straight microchannel with a 90 mT magnetic field, a serpentine channel, and a straight microchannel without a magnetic field. As can be seen, over time the mixing efficiency increases in a specific droplet through the microchannel. According to the results, in the presence of the magnetic field, the mixing efficiency inside droplets increases rapidly. On the other hand, in the absence of the magnetic field, other parameters such as the geometry of the channel and fluid properties affect flow inside droplets as well as mixing efficiency. In the serpentine microchannel, the curvature of channels causes the flow inside droplets, strengthening convection.

## 4. Conclusions

In this study, we numerically investigated the effect of a non-uniform magnetic field on the efficiency of the mixing process within droplets. It was shown that when an external magnetic field was applied perpendicular to the interface of the two solutions in droplets, the mixing efficiency increased more than 15% because of changing of the velocity field within droplets from symmetry to asymmetry. The effect of the concentration of the ferrofluid was also investigated. It was shown that under a specific concentration (0.6 mol/m^3^), the mixing efficiency increased by increasing the concentration of the nanoparticles. However, after that, the mixing index did not change noticeably. The main reason for that is that the interface between the ferrofluid solution and water is limited and does not increase by increasing the ferrofluid’s concentration (it does not have additional convection effects). In addition, by investigating the effect of the magnetic field’s intensity, it was shown that as the magnetic field intensifies, the magnetic force makes the circulation flow inside the droplet asymmetric, leading to a chaotic advection, which creates a flow inside the droplet that increases the mixing efficiency dramatically. Furthermore, the comparison between passive and active methods showed that the use of a magnetic field enhanced the mixing efficiency by more than 10 percent compared to the serpentine platform. This method is a practical solution for rapid mixing within droplets without changing the pH level, ion concentration of the fluids, or designing complex geometries [43]. Therefore, this method does not have negative side effects of using acidic or basic pH on materials or organisms encapsulated in droplets.

## Figures and Tables

**Figure 1 micromachines-13-01661-f001:**
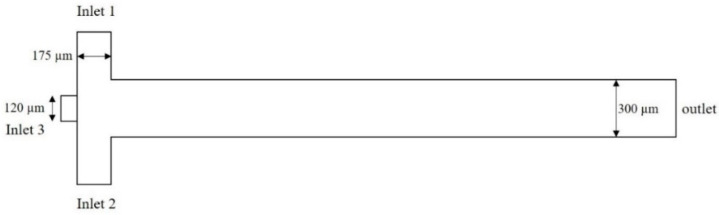
The 2D overview of droplet generation channel and the placement of the magnet.

**Figure 2 micromachines-13-01661-f002:**
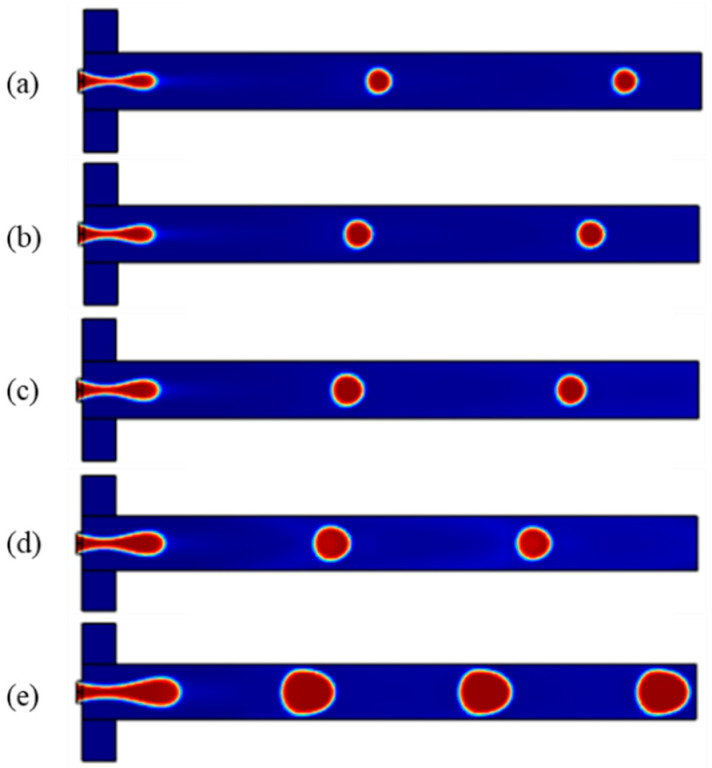
Droplet formation in capillary number = 0.137 and different flow ratios: (**a**) 0.083, (**b**) 0.1, (**c**) 0.125, (**d**) 0.167, (**e**) 0.25.

**Figure 3 micromachines-13-01661-f003:**
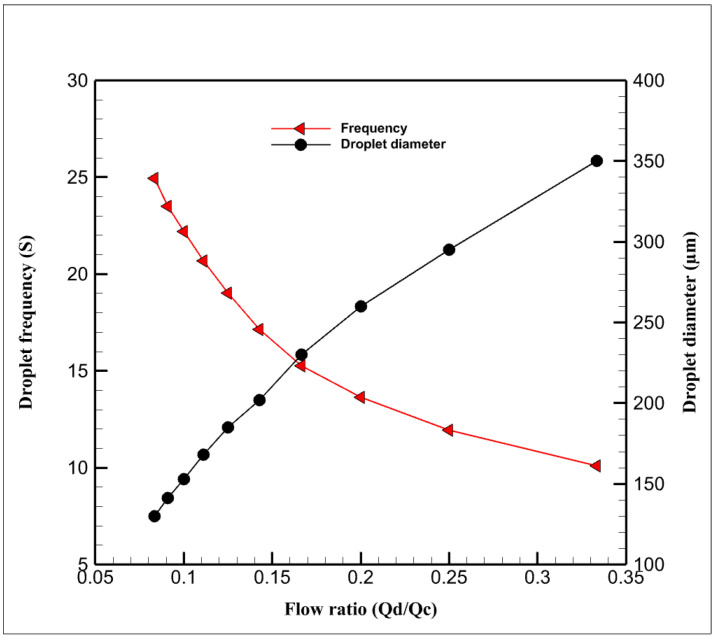
Droplet diameter and droplet frequency as functions of flow ratio.

**Figure 4 micromachines-13-01661-f004:**
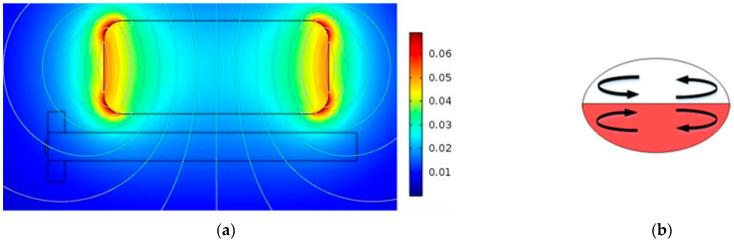
(**a**) Magnetic field parallel to the microchannel, (**b**) the effect of the created convection on the mixing process.

**Figure 5 micromachines-13-01661-f005:**
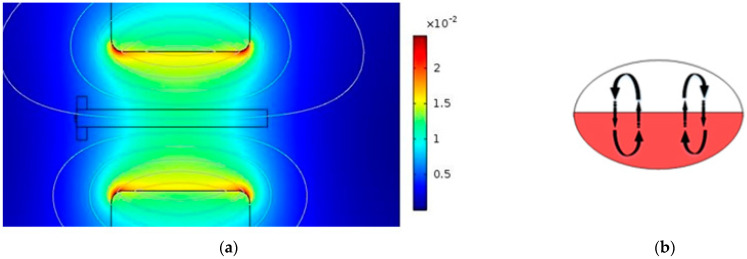
(**a**) Magnetic field perpendicular to the microchannel, (**b**) the effect of the created convection on the mixing process.

**Figure 6 micromachines-13-01661-f006:**
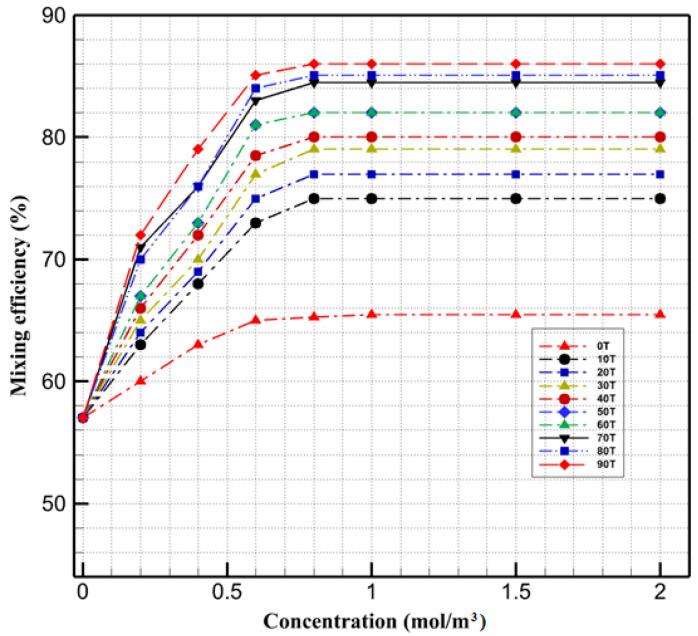
Mixing efficiency in different concentrations of magnetic nanoparticles and magnetic field intensities.

**Figure 7 micromachines-13-01661-f007:**
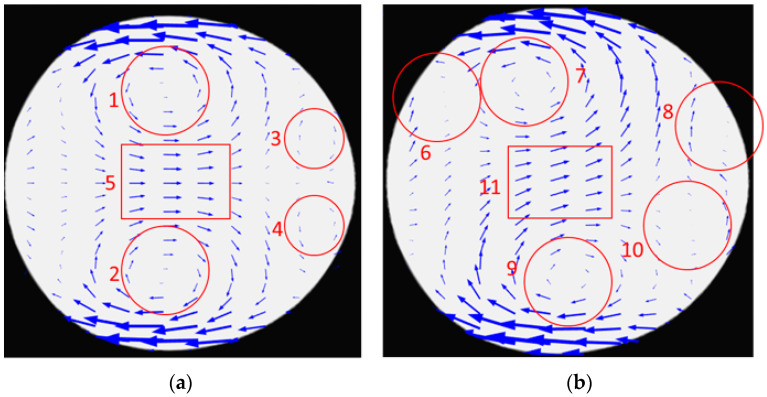
Velocity field inside droplets: (**a**) symmetric velocity field in the absence of the magnetic field, (**b**) asymmetric distribution in the presence of magnetic field.

**Figure 8 micromachines-13-01661-f008:**
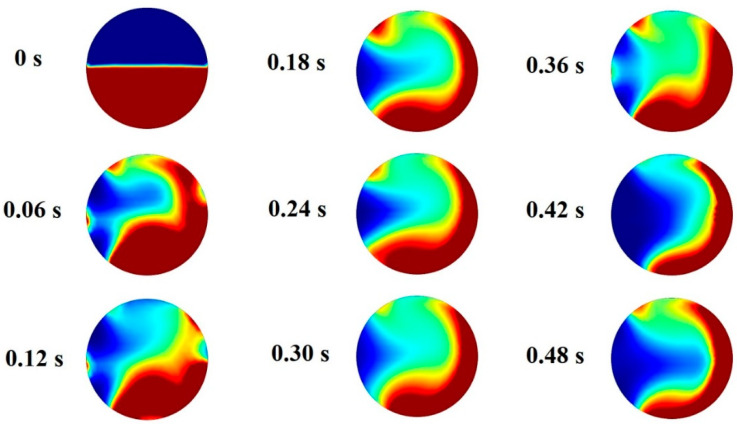
Concentration of the ferrofluids inside droplets in various stages.

**Figure 9 micromachines-13-01661-f009:**
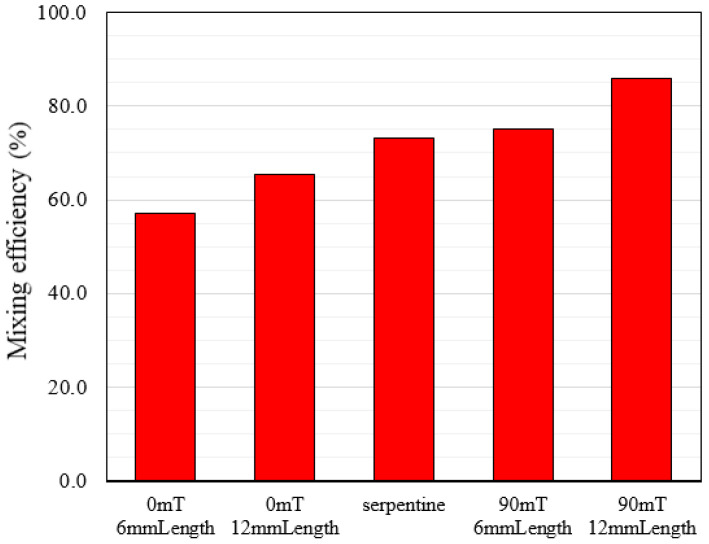
Mixing efficiency in active and passive methods.

**Figure 10 micromachines-13-01661-f010:**
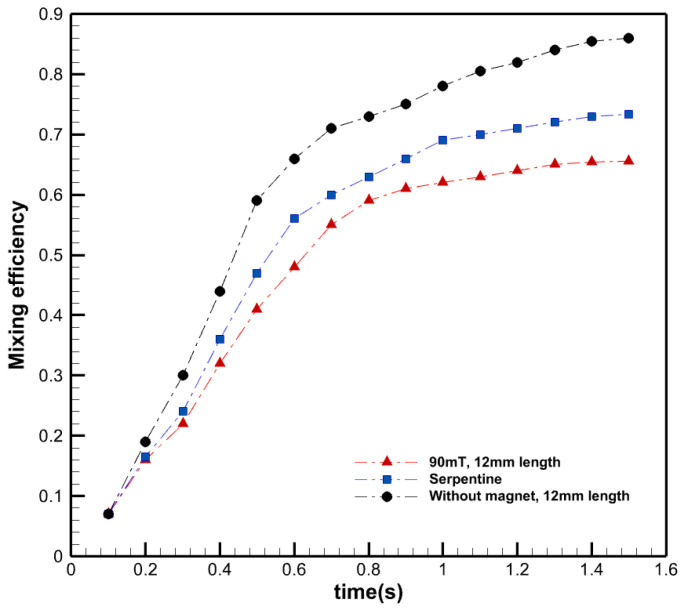
Mixing efficiency in different time steps in active and passive methods.

**Table 1 micromachines-13-01661-t001:** Validation parameters.

Results	Droplet Size (µm)	Mixing Efficiency
Our simulation	208	46.1%
Bai et al. [41]	240	47%
Error	13.3%	1.9%

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
