# Peer review of "The Effect of Non-Uniform Magnetic Field on the Efficiency of Mixing in Droplet-Based Microfluidics: A Numerical Investigation"

_micromachines, 2022, doi:10.3390/mi13101661_

Round 1

Reviewer 1 Report

Rezaeian et al. presented "The effect of non-uniform magnetic field on the efficiency of mixing in droplet-based microfluidics: A numerical investigation". they studied the influence of the magnetic field parameters on mixing performance inside droplets. There is no substantial change in mixing efficiency when varying concentrations of ferrofluids are subjected to a constant magnetic field. As the magnetic field strengthens, the magnetic force causes the circulation flow inside the droplet to become asymmetric, resulting in chaotic advection and flow, which boosts the mixing efficiency. They discovered that using magnetic fields to improve mixing efficiency inside droplets boosts mixing efficiency from 65.4% to 86.1% by increasing the magnetic field strength from 0 to 90mT. Aside from that, the influence of ferrofluid concentration on mixing efficiency is investigated. It is demonstrated that increasing the ferrofluid concentration from 0 to 0.6 mol/m3 significantly improves mixing efficiency.  Overall, the manuscript is well written, and the results supports the claims of the authors. I have the following minor comments:

*More of the related works should be included from the resent literature in the introduction.

*Did the authors consider mixing time in this work?

*Units should be included in table 1 for droplet size.

*What would be the effect of the fluid viscosity to mixing behavior?

Author Response

Reviewer 1:

Rezaeian et al. presented "The effect of non-uniform magnetic field on the efficiency of mixing in droplet-based microfluidics: A numerical investigation". they studied the influence of the magnetic field parameters on mixing performance inside droplets. There is no substantial change in mixing efficiency when varying concentrations of ferrofluids are subjected to a constant magnetic field. As the magnetic field strengthens, the magnetic force causes the circulation flow inside the droplet to become asymmetric, resulting in chaotic advection and flow, which boosts the mixing efficiency. They discovered that using magnetic fields to improve mixing efficiency inside droplets boosts mixing efficiency from 65.4% to 86.1% by increasing the magnetic field strength from 0 to 90mT. Aside from that, the influence of ferrofluid concentration on mixing efficiency is investigated. It is demonstrated that increasing the ferrofluid concentration from 0 to 0.6 mol/m3 significantly improves mixing efficiency.  Overall, the manuscript is well written, and the results supports the claims of the authors. I have the following minor comments:

*More of the related works should be included from the resent literature in the introduction.

 Thank you for the suggestion. Recent studies are reflected in the updated manuscript (Page 16).

*Did the authors consider mixing time in this work?

As it has been shown in figure10, the mixing efficiency increase continuously as the time passes.

Thus, the mixing efficiency should be measure in a specific time. In this study, the mixing time was 1.5 s.

*Units should be included in table 1 for droplet size.

 The Unit of the droplet’s size in table 1 has been added in the revised version.

*What would be the effect of the fluid viscosity to mixing behavior?

For viscous fluids, the diffusivity is lower which increases mixing time [1]. Furthermore, the fluid viscosity can affect the mixing by changing the movements of fluid’s layers in the convection process. Therefore, when the viscosity increases, the mixing efficiency reduces in a specific time interval.

References:

[1]        J. D. Tice, A. D. Lyon, and R. F. Ismagilov, “Effects of viscosity on droplet formation and mixing in microfluidic channels,” Anal. Chim. Acta, vol. 507, no. 1, pp. 73–77, Apr. 2004.

Reviewer 2 Report

The article entitled "The effect of non-uniform magnetic field on the efficiency of mixing in droplet-based microfluidics: A numerical investigation" presents the magnetic field-based strategies for enhancing the mixing efficiency inside droplets. The authors describe the detailed system conditions and results from recent studies and provide insight into the need for achieving high efficiency and throughput in droplet-based mixing. Moreover, the manuscript is well-written and -referenced.

Major comments:

1. It is recommended for the authors to discuss any potential limitations using the suggested models including assumptions and boundary conditions and applicability to other material parameters and geometries.

2. The authors mentioned two different mixing behaviors depending on the intensity of magnetic fields applied. It is suggested to provide more clear evidence of how these two behaviors could be shifted.

3. In addition, what happens when the two different mixing behaviors co-exist, and which one would be more dominant at the intermediate intensities?

4. It is recommended for the authors to provide some potential experimental/simulational designs to achieve high efficiency and throughput in droplet-based mixing.

Author Response

Reviewer 2:

Comments and Suggestions for Authors

The article entitled "The effect of non-uniform magnetic field on the efficiency of mixing in droplet-based microfluidics: A numerical investigation" presents the magnetic field-based strategies for enhancing the mixing efficiency inside droplets. The authors describe the detailed system conditions and results from recent studies and provide insight into the need for achieving high efficiency and throughput in droplet-based mixing. Moreover, the manuscript is well-written and -referenced.

Major comments:

  1. It is recommended for the authors to discuss any potential limitations using the suggested models including assumptions and boundary conditions and applicability to other material parameters and geometries.

Thank you for the recommendation. As it is mentioned in the manuscript, efficient mixing in microfluidic devices is broadly used in bio applications such as cell lysis and Polymerase Chain Reaction (PCR). Stability and biocompatibility are two issues regarding the application nanoparticles which have been investigated in literatures [2]. The materials and boundary conditions have been discussed on page 4 and page 7 in the updated manuscript.

  1. The authors mentioned two different mixing behaviors depending on the intensity of magnetic fields applied. It is suggested to provide more clear evidence of how these two behaviors could be shifted.

 Shear forces, stemming from viscous tension stresses applied by a thin film of continuous phase on the droplet, give rise to velocity difference between the droplets and mean flow in the channel. It will end up making a symmetric internal recirculation flow within the droplets playing a crucial role to mix two miscible phases. At low magnetic fields, due to the magnetic susceptibility gradient at the interface of two fluids, magnetic nanoparticles transport from high concentrated regions to those low concentrated slowly. In this state, it can be turned out that diffusion is the dominant mode of mixing process inside the droplets. When being adequately large, magnetic field induces dominant magnetic force disturbing the internal recirculation flow, producing vortices and flow inside droplets. This way, nanoparticles’ transportation, prompted by dominant convection flow, enhances mixing inside the droplets dramatically.

  1. In addition, what happens when the two different mixing behaviors co-exist, and which one would be more dominant at the intermediate intensities?

At very weak magnetic fields, molecular diffusion is the dominant mode of droplet-based mixing. At the intermediate intensities, when two mechanisms co-exist and proceed mixing behavior, mixing inside the droplets increase broadly by convection bulk flow, and the effect of diffusion would be negligible. Thus, molecular diffusion would only slightly improve the mixing efficiency.

  1. It is recommended for the authors to provide some potential experimental/simulational designs to achieve high efficiency and throughput in droplet-based mixing.

Hitherto, several studies have been conducted to establish the ability of passive methods enlisting dean vortices generated by centrifugal nature of curved channels to enhance the mixing efficiency [3, 4]. For example, serpentine sio-suidal channel with contraction-expansion sections have been proved to hasten and increase the mixing inside the droplets. Moreover, in this study, we have shown the ability of non-uniform external magnetic field to enhance the droplet-based mixing significantly. Taking all the aforementioned hints into account, we can consider a combination of passive and active methods in which serpentine microchannel with contraction-expansion sections integrates with a non-uniform external magnetic field, as a potential concept to achieve high mixing efficiency. In addition, Expanding the channel length in initial U-trurns of serpentine would increase the mixing length and stand the mixing efficiency in good stead. 

References:

[1]        J. D. Tice, A. D. Lyon, and R. F. Ismagilov, “Effects of viscosity on droplet formation and mixing in microfluidic channels,” Anal. Chim. Acta, vol. 507, no. 1, pp. 73–77, Apr. 2004.

[2]        S. Genc and B. Derin, “Synthesis and rheology of ferrofluids: a review,” Curr. Opin. Chem. Eng., vol. 3, pp. 118–124, Feb. 2014.

[3]        Q. Yu, X. Chen, X. Li, D. Z.-C. in H. and M. Transfer, and  undefined 2022, “Optimized design of droplet micro-mixer with sinusoidal structure based on Pareto genetic algorithm,” Elsevier.

[4]        Y. Fu et al., “Simulation of reactive mixing behaviors inside micro-droplets by a lattice Boltzmann method,” Elsevier.
